# A qualitative study of bereavement support volunteers' views and experiences on an online Acceptance and commitment therapy-based (ACT) training programme

Anne Canny[1]*, David Gillanders[1], Tamzin Burnett[2], Brooke Swash[3], Juliet Spiller[2], Emily Harrop[4], Lucy Selman[5], Nicola Reed[6], Anne Finucane[1,2]

1 Clinical Psychology, School of Health in Social Science, University of Edinburgh, United Kingdom, 2 Marie Curie Hospice, Edinburgh, United Kingdom, 3 Division of Psychology, University of Chester, Chester, United Kingdom, 4 Marie Curie Research Centre, Division of Population Medicine, School of Medicine, Cardiff University, United Kingdom, 5 Palliative and End of Life Care Research Group, Population Health Sciences, Bristol Medical School, University of Bristol, Bristol, United Kingdom, 6 Cruse Scotland, Edinburgh, United Kingdom

* anne.canny@ed.ac.uk

## Abstract

### Background

Grief is a natural process, and many people will adjust in time with support from family and friends. However, evidence suggests that around 40% of bereaved people may benefit from additional assistance, including support from bereavement volunteers. Despite the recognition that bereavement care is a public health priority, availability of bereavement support is inconsistent across the UK and internationally. The continuing expansion of online connectivity offers opportunities to develop digital health interventions to help address the needs of grieving individuals. To improve access to bereavement support, we developed an online intervention based on Acceptance and Commitment therapy-based Training (ACT) 'My Grief My Way' and trained volunteers to provide bereavement support in line with ACT-based principles.

### Aim

To describe the views and experiences of bereavement support volunteers who undertook online ACT-based bereavement support training designed to help bereaved individuals cope with grief and improve quality of life.

### Design

Semi-structured interviews and focus groups were conducted with a convenience sample of bereavement support volunteers from two not-for-profit bereavement services in UK. Analysis was guided by the framework approach via NVivo-14.

**Data availability statement:** The de-identified data underlying the results presented in the study are available at reasonable request from the ethics committee at the School of Health in Social Science at the University of Edinburgh: ethics.hiss@ed.ac.uk.

**Funding:** This research was funded by a Research Project Grant from Marie Curie, Ref: MC-21-808. AF is funded by a Marie Curie Research Fellowship.

**Competing interests:** The authors have declared that no competing interests exist.

## Results

A total of 17 participants were recruited; age range 33–76 years, female, n = 15 (88%); ethnicity white, n = 17 (100%). Of these, 15 completed ACT-based My Grief My Way training. Nine participants took part in two focus groups (n = 7) or individual interviews (n = 2), Training was perceived positively, with resulting themes and subthemes indicating there was something to suit everyone's learning preferences. Participants described the benefits of incorporating ACT-based principles and strategies as valuable additional tools to current practice, underlining the model's relevance, compatibility and practical application, and was viewed as a good fit irrespective of which therapeutic approach they used with clients. Online ACT-based training and the delivery of ACT-based bereavement support was therefore, perceived as a valuable approach in this context.

## Introduction

### Background

Grief is commonly understood as a natural and inevitable part of human existence. How individuals respond to this normal but painful experience varies greatly and can involve an extensive range of cognitive, emotional and physical reactions.[1] The public health model of bereavement describes a three-tiered approach to bereavement risk and support needs [2,3]. Most people (60%) will experience normal grief reactions and will adjust in time with support from family, friends and social networks (tier one – low risk) [4]. However, for 30% of people, additional support may be required (tier two – moderate risk). A further c.10% are at risk of developing complicated grief disorder, (also known as prolonged grief) [5] requiring targeted professional mental health services (tier three – high risk) [2,3].

Bereavement support is a central component of health and social care strategies in many countries [6–11] and increasingly recognised as a public health priority [8]. Yet, despite this acknowledgement, bereavement care is lacking across locations and settings [12–14]. The UK National Health Service (NHS) offers guidance and provides bereavement support via signposting to bereavement charities and other third sector organisations which, through a range of qualified counsellors or trained volunteers, provide support to those in need [7,15]. Volunteer support is widely recognised as a valuable resource to aid bereaved individuals [15–18]. however, over-stretched services with lengthy waiting times for appointments, poor communication systems and a lack of information on clear pathways to help has led to much unmet need [14]. Greater availability and timely, straightforward access to evidence-based services are therefore of utmost importance [19,20].

To increase availability and timely access to bereavement support, we developed an online intervention built around Acceptance and Commitment Training (pronounced as one word, ACT rather than three letters)) [21]. ACT is part of the 'third wave' of behaviour therapy and aims to reduce suffering and increase wellbeing via

three core overlapping and interdependent processes of change [21]. These can be translated as Psychological Flexibility, comprising: The ability to be AWARE – in the present moment; to be OPEN – accepting of difficult feelings and emotions; and to be ENGAGED – to move forward in life by doing what matters [22] (Fig 1).

ACT is well established in the treatment of common mental health problems such as anxiety, depression, substance use and chronic pain [24]. However, evidence of its efficacy for bereavement is limited. A recent study suggested ACT helped reduce distress and anticipatory grief for people living with advanced progressive illness, their caregivers and staff involved in their care [25]. A further study found that parentally bereaved college students who were more experientially avoidant (psychologically inflexible) were more likely to develop complicated grief, indicating psychological flexibility as the mechanism for behaviour change in bereavement [26]. In addition, Davis and colleagues demonstrated acceptance and valued-living practices, key components of ACT-based principles helped in adjustment to grief in bereaved Australian students [27]. Finally, our team previously found that ACT practitioners described improved coping and wellbeing in clients following ACT-based bereavement support [28].

ACT is a pragmatic, structured yet flexible approach. The model is compatible with other established grief theories such as the Dual Process Model of bereavement (DPM) [29]. Continuing Bonds [30]., and Meaning Making approaches [31]. For example, the DPM posits that bereaved people oscillate between dealing with the loss of the deceased person (loss-orientated coping (LO) e.g., grief work) and negotiating the practical and psycho-social changes to their lives that occur as a result of the bereavement (restoration-orientated (RO) coping, e.g., forming new roles/ identities/relationships) [32]. ACT can add to this by providing empirically supported intervention techniques that help people to engage with painful loss-related emotions (loss-oriented coping) and to take perspective, build meaning and engagement in life after bereavement (restoration-oriented coping).

This study was part of a wider project which developed an online ACT-based intervention to improve ability to cope and quality of life after bereavement, (ISRCTN18357870 https://doi.org). In summary, this 24-month research consisted of two phases [33,34]. Phase one included the development of a programme theory and logic model stemming from evidence of bereavement theories from existing literature [28,35]. (S7 and S8 File) Phase two comprised the development of an online bereavement support intervention prototype website 'My Grief My Way' (MGMW -https://mygriefmyway.co.uk) which could be accessed by bereaved individuals in one of two ways; self-directed or volunteer-supported. The prototype website was tested for both approaches then refined iteratively using a mixed methods design [36]. This article describes bereavement support volunteer views on the associated ACT-based training programme and their experiences of supporting bereaved clients using MGMW in addition to their usual bereavement support sessions.

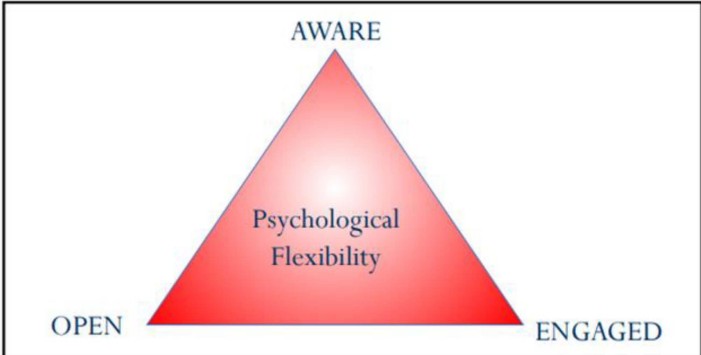

**Fig 1. Psychological Flexibility within Acceptance and Commitment Therapy (ACT).** [23].

**Study aims**

We aimed to explore bereavement support volunteer perspectives and experiences of an ACT training programme based on the following research questions:

a. What are the views of bereavement support volunteers regarding training in ACT for bereavement support?

b. What are the experiences of bereavement support volunteers regarding the delivery of ACT-based bereavement support (through MGMW) to bereaved individuals?

c. What components of MGMW are useful in supporting the wellbeing of people with bereavement support needs?

## Methods

### 2.1. Design

This study used qualitative methods encompassing semi-structured interviews and focus groups to allow a rich understanding of participant perspectives, sensitive to the context of the study [37]. The Consolidated Criteria for Reporting Qualitative research (COREQ) was used to guide reporting [38].

### 2.2. Participants and recruitment procedure

Participants were bereavement support volunteers from two national not-for-profit organisations which offer no-cost bereavement support counselling and listening services to bereaved individuals in the United Kingdom. Volunteers were either qualified counsellors or trained by respective charities to deliver bereavement support in accordance with national guidelines [15]. A volunteer was then matched with a bereaved person to provide bereavement support over the telephone, online and sometimes in person for up to six sessions (and sometimes more if required), each lasting approximately 45 minutes. Bereavement services were accessed through free of charge telephone helplines where individuals could talk in confidence about their grief.

Managers at respective organisations shared the Participant Information Sheet via team meetings, email bulletins and organisational communication platforms. (S6 File) Bereavement support volunteers were eligible to take part if they met the following criteria: (a) had completed respective organisational bereavement support training; (b) had been actively supporting bereaved clients for at least one year; and (c) spoke fluent English. Student counsellors who are often put on placement to gain hands-on experience in real-world healthcare settings within stakeholder organisations were excluded for continuity purposes. Recruitment began on 21st November 2023 and ended on 29th December 2023.

We aimed to recruit 20 bereavement support volunteers by means of convenience sampling [39]. The volunteer sample size was based primarily on pragmatic considerations, i.e., the number of volunteers potentially available in consultation with key stakeholder organisations [40].

Interested potential participants contacted their respective manager and gave verbal consent for the research team to make contact. A member of the research team (AC) then provided an online consent form for each participant to complete. Once consented, participants were assigned a unique six-digit ID to protect identity and invited to complete a short background questionnaire so that the study sample could be described (e.g., organisation affiliation, age, gender, ethnicity, years' experience with organisation and level of skill as a volunteer).

### 2.3. ACT training for bereavement support volunteers

Volunteer participants took part in four training sessions on the principles of ACT, each lasting two hours over a period of one month. Training sessions were delivered via Microsoft Teams by the co-project lead (DG) who is a Clinical Psychologist and has extensive training and experience in ACT. DG is also a Peer Reviewed ACT Trainer and Fellow of the Association for Contextual Behavioural Science, an indicator of quality and fidelity in ACT knowledge and skill.

ACT training sessions covered the basic theory of psychological flexibility, a key concept in the ACT model [23,41]. Training covered typical application strategies and specific tools for using ACT with people who are grieving. (S1–S4 File) The training format included experiential exercises, ideas and practices alongside group discussions to offer additional ACT-based support to bereaved individuals using the MGMW website. (Table 1).

On completion of training, managers at each not-for-profit organisation matched volunteers with bereaved individuals in line with their respective organisational procedures. In addition to the four training sessions, bereavement support volunteers were invited to group supervision meetings facilitated by DG once they started providing ACT-based bereavement support. A total of four 90-minute supervision meetings were held, each being unstructured and led by volunteers. These meetings, delivered online via Microsoft Teams, explored volunteer experiences of drawing on ACT principles when supporting bereaved individuals, alongside troubleshooting any issues that came to light, enhancing therapist skills and flexibility.

## 2.4. Data collection

Participants were invited to take part in either a focus group or a one-to-one interview (depending on participant preference). Focus groups were conducted via Microsoft Teams by two female researchers with experience in palliative care and bereavement; study research associate AC, and TB, a qualified counsellor of 15 years who undertook qualitative research training on conducting focus groups and framework analysis prior to conducting the focus groups. The researchers were unknown to the participants prior to the study.

A semi-structured interview topic guide was developed and reviewed by our Patient and Public Involvement (PPI) group (9 bereaved individuals) and the study steering group. PPI members and our steering group were asked to pay particular attention to reliability and validity to ensure the topic guide reflected study aims to offer a focused yet flexible approach to capture details that were salient to each individual participant [42]. (S5 File) Focus groups were audio and video recorded via Microsoft Teams whilst individual telephone interviews were audio recorded on a password protected digital recorder. Recordings were transcribed verbatim and anonymised by AC, prior to analysis in NVivo 14.

**Table 1. Bereavement support volunteer training summary schedule.**

| Bereavement Support Volunteer Training Schedule |
| --- |
| **a. Session 1: What is ACT and why ACT for grief?** |
| a. Training covered the six processes of the ACT model (Acceptance, Cognitive Defusion, Being Present, Self as Context, Values, Committed Action), through three broad skill sets (Aware, Open, Engaged). How do problems in these skills lead people to be stuck in their grief? |
| b. Common sticking points in responding to loss. Seeing problems with grief through an ACT lens; seeing grief as natural healing; examples of ways that ACT techniques can unblock the natural grieving process. |
| **Session 2: Developing Awareness and Openness Skills.** |
| a. Language around acceptance, track behaviour patterns and factors of influence. Where will that lead? Learning to let go and allow the pain of grieving. To develop self-compassion and patience. Learning to trust the process. |
| b. When your own mind is not your friend, how to change your relationship with your own thoughts; catch the old story and begin to write a new chapter. |
| **Session 3: Developing Engaged Skills and Reconnecting.** |
| a. Continuing bonds with the deceased and bring them with you into a new chapter. |
| b. Have courage to try new things, willingness to try and fail, being open to the new and unfamiliar. |
| **Session 4: Supporting People with My Grief May Way.** |
| a. Starting from where you are at – ensuring a good fit between ACT and existing skills and frameworks. |
| b. Supporting choices, trusting processes, troubleshooting frequent problems. |

## 2.5. Data analyses

Data analyses were informed by the framework method [37], a flexible analytical process that supports key steps in data management [43,44] and suitable for multi-disciplinary health research teams [43,45]. Guided by the framework approach, researchers AC, AF and BS independently coded two identical transcripts (one focus group and one individual interview) using a combination of inductive coding (ground-up from the data) and deductive coding (guided by the research questions). Initial themes were organised into a set of categories to form a series of refined category matrices [45]. Through in-depth team discussions, consensus was reached on final themes, to which the remaining study data were coded, resulting in systematically structured outputs of summarised data [45].

## Ethics statement and consent to participate

All procedures described in this study were reviewed and written approval was received by the Clinical Psychology Ethics Committee of the School of Health in Social Science at the University of Edinburgh. (Ref: CAHSS2309/02) and by the Research Governance Team of Marie Curie (Ref: 23MC008). All bereavement support volunteers in this study provided written informed consent to participate.

# Results

## 3.1. Participant characteristics

A total of 17 bereavement support volunteers took part in the ACT training programme. Of these, one participant withdrew during training and a further two participants withdrew after training. All withdrawals were due to other competing work-related priorities. One participant submitted their background information questionnaire without their participant ID therefore; we could not attribute details to a specific participant. Age range of participants, 33–76 years, female, n = 15 (88%); ethnicity white, n = 17 (100%). Details are shown in Table 2.

## 3.2. Participation in ACT training

A total of 15 participants completed ACT Training sessions and of these, 11 were matched with a bereaved individual. Those unmatched remained on a waiting list to be paired with a bereaved individual, however, no suitable matches had been identified by the time the data collection had finished. Six volunteers were matched with more than one bereaved client as shown in Table 2. Overall, nine participants took part in focus groups or individual interviews (60% of those who completed training). Focus groups each lasted approximately 1.5 hours whilst individual interviews lasted around 40 minutes.

Analyses of interviews and focus groups with 9 participants resulted in four key themes with sub-themes as shown in Table 3.

**3.2.1. Training structure.** Length of course: Most participants referred to the length of four, two-hour sessions over a period of four weeks as receiving a good balance of structure, content and personal time commitment, but one participant would have like some hands-one training or additional content. Two participants had previous knowledge of ACT, making the learning experience more familiar.

> "I possibly would have suggested a little bit longer and maybe some practical sessions. Just some triad work (role play) and I guess maybe it's a reflection of the sort of learner that I am." (BSV215)

All participants acknowledged the convenience online learning afforded, referencing time and cost savings associated with travelling. Participants also mentioned the aftermath of Covid-19 pandemic where most people made the shift to work remotely and engaged with new forms of technology, making the online nature of training more acceptable.

Table 2. Characteristics of recruited bereavement support volunteers.

| | Bereavement Support Volunteer Participant Characteristics | | | | | | | |
|---|---|---|---|---|---|---|---|---|
| ID | Organisation 1 or 2 | Age | Gender | Length of volunteer service (years) | Level of skill | Number of clients supported post-training | Attended Focus Group | Attended Individual Interview |
| BSV101 | 1 | 36 | Female | More than 5 | NICE level 2 [a] | 0 | | |
| BSV102 | 1 | 59 | Female | More than 5 | NICE level 2 [a] | 0 | | |
| BSV103 | 1 | 63 | Female | 4-5 | Other [a] | Withdrew after training | | |
| BSV104 | 1 | 33 | Female | 1-2 | NICE level 2 [a] | 3 | | |
| BSV105 | 1 | 70 | Female | 3-4 | NICE level 2 [a] | 2 | ✓* | |
| BSV107 | 1 | 69 | Female | 1 | Other [a] | 3 | | |
| BSV211 | 2 | 73 | Female | More than 5 | Counsellor | 1 | ✓ | |
| BSV212 | 2 | 64 | Female | 1 | Counsellor | 1 | ✓ | |
| BSV213 | 2 | 65 | Female | More than 5 | Advanced listener [a] | 2 | | ✓** |
| BSV214 | 2 | 76 | Female | More than 5 | Advanced listener [a] | 1 | ✓ | |
| BSV215 | 2 | 69 | Female | More than 5 | Skilled listener [a] | 1 | | ✓ |
| BSV216 | 2 | 56 | Female | 2-3 | Counsellor | 3 | ✓ | |
| BSV217 | 2 | 49 | Female | 1-2 | Counsellor | Withdrew during training | | |
| BSV218 | 2 | 62 | Male | 4-5 | Counsellor | 1 | ✓ | |
| BSV219 | 2 | 48 | Female | 3-4 | Counsellor | 0 | ✓ | |
| BSV220 | 2 | 65 | Female | More than 5 | Counsellor | 2 | | |
| Missing ID [b] | 1 | 58 | Male | 3-4 | NICE level 2 [a] | Withdrew during training | | |

[a]. NICE (National Institute for Health and Care Excellence). Different levels of skill correspond to the UK NHS National Bereavement Care Pathway bereavement support training needs.[3,15].

[b]. Table 2 shows 17 participants, including data from one participant that was submitted without a participant ID and therefore, could not be assigned to an individual ID.

**\*** Attended x 2 focus groups. **\*\*** Attended x 2 interviews.

"I guess you know, we're all getting pretty familiar with online…. I was always most likely to be seeing my client on video. And there's something about this whole setup which has a sort of 'onliney' feel about it. That makes it more natural to do the training online you know? Because you can always imagine like if you're doing it in person, you'd have to be sitting there in the room imagining what it was like being on the website and stuff." (BSV218)

Materials and content: Participants were positive about the training materials and appreciated the variety of teaching approaches, including oral and visual presentations that would appeal to a wide range of learning styles. The use of metaphors and analogies, often used in ACT to assist with making abstract concepts concrete were reported to complement examples of ideas and practices, supporting student learning.

"I thought the discussions and the material were good and it helped me understand what ACT was really all about. And I quite liked whenever we had the audio, the relaxation (mindfulness). So, we were actually experiencing what clients could experience through the website on the audio relaxation tips etcetera." (BSV213)

Handouts were considered very helpful, as was being able to access video recordings of training sessions via a Teams link that was sent to all participants following every session.

"And practically, I liked the ability to revisit the materials that were recorded online, and I found myself, you know, if I was trying to just refresh myself in something to take myself back to the recorded training materials." (BSV219)

**Table 3. Key themes and sub-themes from focus groups and interviews on ACT training and delivery.**

**Bereavement Support Volunteer – ACT Training and Delivery**
**Key Themes and Sub-themes**

| Key themes | Sub-themes |
|---|---|
| Training structure | |
| | Length of course |
| | Materials/content |
| | Understanding ACT |
| | ACT Value |
| | Training suggestions |
| Acceptability | |
| | Website accessibility/flexibility |
| | Website content/materials |
| | Website suggestions |
| | Confidence using ACT in practice |
| | Client engagement/benefit/trust |
| Training Sustainability | |
| | Preserving training quality |
| | Level of trainee skill set |
| Moving forward with the delivery of ACT-based bereavement support | |
| | Continued access to MGMW |
| | Planned future use |

Every participant commented on the teaching style by expressing how sessions were delivered in an engaging, personable and therapeutic manner.

"I thought (name of facilitator) was very engaging and so it was very easy to listen to him and very helpful, very informative. And I really liked his responses to questions that came up in the group. He would always be very thoughtful, and he would really elaborate and just flesh things out. He would make everything three-dimensional in a really lovely way. So, I found that really helpful." (BSV212)

Understanding ACT: Most participants considered the ACT model easy to understand and were able to identify with ACT principles, ideas and practices.

"I think, you know, because I was making notes and stuff was there. It was easy to understand. It was emphasized, you know, having the slides and talking about it and stories and I think it was taught well. I understood it. It, yeah, it did the job… It sort of brought out, made me more aware of what I'm doing." (BSV105).

One participant however, felt the need for the facilitator to elaborate further regarding the Aware, Open, and Engaged components of the training. She experienced being puzzled as to how one section led to another and would have liked more input on why this particular approach is of value for bereavement.

'I just felt there was a bit of a leap. So, we are picking up on the aware, open, engaged part of the thing. How did you quite get us there? Ohh, alright, we're here then, and that's ideal for bereavement. OK. But what I could have done with was just a wee bit more explanation as to why that had been particularly the place to go to and I think no doubt it was the right place, but it was just that sort of logical little leap." (BSV214)

ACT Value: Many participants highlighted how much they valued ACT training and how the concept seemed to validate and enhance their understanding of their own practice. ACT was felt to be congruent with existing ways of working, which functioned to provide a structure to how existing practice could be understood and extended using ACT tools, strategies and MGMW.

"I think it is about sort of like recognising things that you're already doing as well. But it's maybe… it's described in a different way… I think I realized that actually most of the time I am doing it (ACT), that's the way that I work. But it was just sort of maybe, you know, the language used." (BSV216)

Participants particularly valued the experiential aspects of the training, which included examples of actually doing ACT and not just simply being taught the model.

"I think (name of facilitator) delivery is great. I love the way that he is very experiential as well… He's, you know, he's led us through some of the practices so that you're actually experiencing it for yourself, which I think is really important when you're then sort of, you know, hoping to use it with clients." (BSV216)

Training suggestions: Handouts for each training session were provided ahead of each training session and included background information on ACT and a list of further reading. (S1-S4 File) Some participants would have preferred to receive this information at the outset, prior to the training programme commencing.

"I think maybe before the course started, if I'd had a little bit more information about the whole ACT and sort of background just so I could get my head round it and also have access to the (MGMW) website." (BSV213)

Some participants suggested that future volunteers might benefit from further details of where resource materials were located on the My grief My Way website. This would assist volunteers to sign-post clients as an interim measure until they became proficient at navigating the resource.

"But maybe something that you know, if you're a training other people to use the website with clients, then maybe some kind of site map, would be helpful." (BSV216)

One participant proposed an additional training session to reflect upon the entire programme and also to further consider ways to bring ACT into personal practice.

"But I would have liked a session for where we maybe discussed you know, just our way in with the clients and so on and so forth. And us thinking more of taking the reins you know, and with clients rather than just absorbing all the information about the ACT, yeah. And so, that sort of felt just a wee bit out in limbo there." (BSV214)

**3.2.2. Acceptability.** Website Accessibility/flexibility: All participants found ACT-based principles helpful in the delivery of bereavement support especially when used in conjunction with MGMW website. Participants found the website easy to access and acceptable.

"I think it's a marvellous resource and it's accessible and it's easy to use. And actually, I can think of people in my life that I could recommend it to now." (BSV215)

Participants also expressed positive views regarding website aesthetics and welcomed the natural colour scheme, described as calming and comforting. In their view, the attractiveness of the website made it more likely that they and their clients would revisit it.

"But when I saw the whole thing, I could see that it was woodland green and trees and ferns and things and so and it turned out to be quite soothing". (BSV214)

Participants believed that the MGMW website offered a sense of independence and autonomy to clients which they viewed important. Many highlighted the fact that some people seeking help prefer to deal with grief in a private manner.

"A lot of people want to do more themselves. They want to be more independent… So, if there's something out there that's user friendly and has materials that are relevant to their situation… It (MGMW) can be enormously supportive. And just watching the videos for so many people I imagine could be, oh my goodness, I'm not alone and that I think it's a great way in just to encourage people and just take a look." (BSV212)

The MGMW website resources were also felt to be flexible and tailored when participants directed clients to appropriate materials. Its personalised style helped participants to guide their clients in an open, structured, but flexible manner.

"Well. I was really pleased to see it (website). I thought it was a really good website. I liked the fact that it was all bite sized. You could dip in because it's so hard when you're grieving to focus and I love the fact that it dipped in and out." (BSV215)

Participants talked about using ACT alongside their usual therapeutic approach which complemented and integrated well with ACT principles.

So, if I'm thinking dual process with a client, I'm thinking I'd like to sign post them to the grief swing (on MGMW website). (BSV219)

Some participants felt that for some clients the website use may prove challenging to use due to low IT skills, lack of means to learn, or lack of access to a mobile phone, laptop, iPad and internet access.

"I have a neighbour particularly, but she's not at all computer literate, so that would be a problem… She hasn't got access to a computer. So, I don't know what I would do with her, but yeah." (BSV215)

Website content/materials: Participants enjoyed the variety of website materials available and felt website content aided the delivery of bereavement support.

"I think you know, has a whole variety of things, it's visual, there's that, you know, the worksheets, etcetera. There's the poetry. Umm, so it's a variety of sort of resources anyway that would suit everyone." (BSV213).

Confidence using ACT in practice: After training, participants described feeling suitably equipped to incorporate ACT principles into their existing practice. They felt well prepared to deliver the model and despite some initial apprehension, managed to navigate some concerns relating to new this therapeutic approach. For some, ACT-based principles reflected how they already practiced in general. For others, the website and training served as a new and related set of tools to bolster the support they provided.

"I feel semi confident because I do use bits of ACT anyway, just noticing and naming sort of thing and you know, considering you know that it's normal to feel a range of emotions and accepting being open to them and letting them sort of flow through that process. So, there are bits of it, you know, thinking of thoughts and feelings as clouds in the sky or leaves on the stream. And those are quite nice things to bring in anyway when it feels appropriate." (BSV212)

Client engagement/benefit/trust: Participants described how they used MGMW with their clients, reporting enthusiastically their perceived views in how MGMW helped them to apply new ideas and practices to help clients with their grief. Most participants reported guiding their clients to website resources considered appropriate to their grief.

"I've used the meditation… So, it's a relaxation and winding down thing. She (the client) finds that very useful and on what she particularly talked about was being, you know, being present, being aware of feelings…. They can see that (real stories), and they go, yeah, and look what happened there. That this isn't just me that is in this situation." (BSV105)

Two participants each had a client who did not engage with the MGMW website, although ACT-based principles were used during bereavement support.

"When I asked her, she said she hadn't looked at it (website). And I think that's just because there's just been so much going on. And I do think with her as well, there's been secondary losses. And so, she's lost her home. And I think one issue maybe just the lack of privacy and space to actually you know to maybe look at materials and you know, and find a quiet place that's hers to actually engage with the material." (BSV212)

According to participants, the core concept of psychological flexibility was key to improving client coping and wellbeing. Participants described the worksheets and exercises they used from the MGMW website to assist clients understand their thoughts and feelings. The concept helped normalise grief with strategies to engage with the presence of distressing thoughts and feelings rather than avoid these.

"Yeah. The emotion wheel (worksheet). She did have a look at that, and she thought ohh, I didn't think about all these different emotions. So, it sort of opened her eyes as to the variety of feelings and emotions that she could have." (BSV213)

It was not only website content that assisted participants when delivering bereavement support, but language use taught in training sessions. The phrase 'what matters to you' helped participants identify client values and encouraged goals and behaviours that were associated with these.

"That question of like, what matters and sort of you know, values has been helpful I think for me, I just think about that with my clients. Because I think a lot of clients you know, they have a lot of anxiety about the future. You know, how am I going to do this? And so often, I feel that you know, we only have six sessions and that is such a small part of their grief journey. And so, to be able to help and support them, to think about what matters, you know, as they leave our support really is you know, I think can be incredibly helpful for them and maybe bring some hope as well and you know." (BSV216)

Participants also reported how MGMW offered a safe psychological space and convenient 24-hour access to support the therapeutic process during and outside client appointments.

"I talked about one of the worksheets specifically and I said I realize worksheets aren't for everybody, but you might find it helpful to look at this and it was. I'll see you in a weeks' time and this is what you could be thinking about during the week… I felt that this (MGMW) was a really strong safety net for people... For my client, and I think for some of my other clients that I can't share it with, they would be using this in between speaking to me" (BSV215)

Website Suggestions: One participant suggested new material such as incorporating bullet points into videos to offer a take home message for each topic to enhance user experience.

"I'm a very visual learner for example, so I would want a wee bit more of drawing things together with bullet points. Cause again, knowing that something was in one of the videos, but I had to listen to them all again. Yes, all of them, to see where was that bit again where (name of facilitator) said such and such? Yep, and it took a bit of finding. And so that was interesting. And she (client) also found that, you know, it took a bit to find." (BSV214)

Other types of mediums were suggested to make the MGMW experience more comprehensive.

"Maybe that would be a nice add-on... A podcast type version of some of the mindfulness/ grounding things so that you don't need to switch on a laptop/website... But can access an audio whilst even in bed." (BSV219)

One participant highlighted the need to be sensitive to language use in website content, particularly since grief sometimes involves a person who was not a 'loved one'.

"Just not to assume everybody is a loved one. Yeah, I've had a number of clients just recently for whom the person who died wasn't a loved one." (BSV215)

Whilst all participants valued MGMW website materials, some important issues were brought to light to help improve resources. These involved the inclusion of younger people to expand the age range in videos, experiences and additional training materials.

"But I just thought OK, where's somebody who's a bit younger. Or somebody in their 20s or 30s. I think the lovely lady whose sister had died, she's maybe in her late 30s, early 40s, but I just thought there seemed to be a chunk of the folk where late middle age. Umm, you know, I just thought for folk accessing in the site." (BSV213)

**3.2.3. Training sustainability.** Preserving training quality: Although every participant described the training as an enjoyable learning experience of good quality, one participant expressed concerns over maintaining the high-quality teaching and materials received to ensure future participant learning, engagement and understanding.

"You know, that's the challenge I would have thought for future trainings. You know, he (name of facilitator) can't do them all. I wouldn't have thought, although you know, I probably would put in a vote for getting him to do them all, you know. But how you move well or how you maintain that quality that the others refer to. The quality of particularly as (name of volunteer) was saying, you know, of being able to kind of think around the content of the material that is the main content of the training and respond to different perspectives and different experiences in the trainee group." (BSV218)

Level of trainee skill set: One participant aired concerns over varying levels of experience within qualified volunteers and the probability of a wide range of knowledge and understanding between future participants. Training will need to consider different levels of skill and experience during further training development.

"What sorts of volunteers or what stage of experience and training volunteers are at, or the support people are at when they get involved in supporting My Grief My Way." (BSV218)

**3.2.4. Moving forward with the delivery of ACT-based bereavement support.** Continued access to MGMW website: Many participants asked about continued access to MGMW for themselves and clients following the study period, demonstrating their interest in and commitment to continued use of website materials.

"She (client) wanted bits in between us speaking and I think it (MGMW) was a real resource for her. She was anxious about losing it when the sessions finished, so I was able to check with you all at the last meeting that we had that she would still have access to it because I think she suddenly thought I'm gonna lose this." (BSV215)

Planned future use of MGMW website: Every participant viewed ACT training and the delivery of ACT as a good fit with their existing bereavement practice irrespective of which therapeutic style they used with clients. ACT was considered a valuable stand-alone model as well as a complementary and enhancing tool kit easily integrated with other grief theories. When used in conjunction with MGMW, participants reported website resources greatly assisted in supporting clients through their grief journey.

"So yeah, and just having the My Grief My Way website was wonderful because that was an actual tool kit for whatever stage the client was at. I was talking about giving them a toolbox, a tool kit in order to move forward… You're not just letting go of them and into the void. You've got a support system for them. And I can see once it's possible to do so, and the project is finished, I would be giving this link to my most of my clients." (BSV215)

## Discussion

This study describes the experiences and perspectives of volunteer bereavement support practitioners who took part in an online ACT-based bereavement support training programme, and who supported clients using ACT principles and MGMW website. Findings suggest ACT-based training and the delivery of bereavement support was noted positively by those who took part, underlining its relevance, compatibility and practical application; features further confirmed in participants' reflections on its positive impact in their own practice and their perception of client benefit.

The growth in the adoption of digital health interventions and an increase in internet connection provide opportunities for educators and students, including those involved in volunteer activities to maximise learning prospects and are widely recognised as providing wide-ranging benefits [46–48]. The online aspect of the current training programme facilitated such advantages. For example, attendance from a wide range of locations across the UK was achieved, enabling flexibility in geographical attendance. As a result, the programme offered the potential to expand availability of bereavement support to clients in the future [14]. Additionally, training incorporated a variety of materials designed to suite various learning styles for students to use 24/7, at their own pace and convenience [49,50].

Formative evaluation is central to good quality training programmes [51] and our participants provided important feedback for course improvements as noted. However, one caveat to bear in mind was the concern to maintain the perceived high-quality materials and teaching delivery in future training. Such an important matter will require ongoing evaluation or future refinement/redesigning to preserve programme quality [50,51]. In a similar vein, the possible wide-ranging bereavement knowledge and experience of future volunteers may have to be taken into consideration during future programme development with the prospect of delivering ACT-based principles in a tiered format. This could take the form of three levels. One; familiarisation with grief and grief support incorporating introductory levels of ACT/MGMW. Two; greater depth training in ACT, similar to the current training programme. Three; targeted towards qualified professionals with background knowledge in counselling/psychotherapy where an ACT/MGMW approach could be used with clients with more complex problems. ACT-based training and the delivery of ACT for bereavement has the potential to support the public health model of grief [3], including problematic grief processes through key elements such as acceptance and valued-living concepts found in this study and is consistent with Davis and colleagues' findings [52]. Programmes could be comprehensively evaluated in a similar manner to this current training programme, taking into account suggestions and proposals from our existing and future participants. Evaluation could also include additional questions specific to the suitability of individually assigned levels of ACT/MGMW courses.

ACT has been used in combination with other therapeutic approaches in the treatment of depression, distress, anxiety and chronic disease with encouraging results [53–56]. ACT and MGMW enhanced existing bereavement practices irrespective of which therapeutic approach participants used to address the multifactorial trajectory of the grieving process and was deemed valuable. In this study, participants voiced how integrating other models such as the DPM [29] did not conflict with ACT principles, but instead, ACT provided concrete strategies through the AWARE, OPEN and ENGAGED elements to support the oscillation between loss-oriented and restoration-oriented concepts of DPM. These data support our previous findings based on the perspectives of ACT practitioners, and highlight that ACT can complement other psychological approaches when used to support individuals who have been bereaved [28].

Mindfulness practices are well established and considered valuable in helping clients pay attention in the present moment, aiding the unfolding of experiences moment by moment, decreasing experiential avoidance and improving well-being [57–59]. In this study, participants perceived an improvement in client psychological flexibility through key strategies; AWARE, OPEN, ENGAGED. Some participants practiced mindfulness techniques with their clients via MGMW or directed them to combinations of mindfulness resources to suit client preferences and were considered reinforcing and valuable aides. Using these three key elements, participants were able to support clients to attend to their internal thoughts; to take a step back from these and question whether these were kind or critical narratives and in their overall best interest. Participants perceived the use of visualising practices and analogies helped clients recognise their thoughts, engage with the presence of distressing thoughts and feelings rather than avoid these which helped to untangle the mire of uncertainty, normalising their bereavement. Clients could explore what was important to them in the here and now and how these values could be translated to action, enabling change from unhealthy grieving behaviours.

The course of the grief trajectory is multifactorial with the mourning process involving many mediators such as social, personality and historical variables as well as the relationship with the deceased, nature of the attachment and type of death [60]. Whilst ACT and the MGMW website were perceived as useful, participants noted the model as valuable for some but not all clients. For instance, two clients who did not engage with MGMW alluded to other external factors such as additional life events or stressors. Such barriers impeded individual ability to connect with the resource rather than having an aversion to the website per se. Concerns from volunteer support participants that some clients may be unable to benefit from MGMW due to a lack of IT skills, equipment or confidence are also important issues. The digital divide is not only confined to this study, but a wider topic for Governments and society as a whole [61]. As such, online interventions are not a panacea for bereavement support and other forms are also needed.

### Strengths and limitations

These preliminary findings of an early intervention development study are the first that we know that evaluate ACT-based training and the subsequent delivery of ACT bereavement support by volunteers. The study therefore builds upon the limited evidence to support ACT-based interventions for bereavement. The richness of volunteer-participant data addresses the evidence gap and offers an understanding in a real-world context in which volunteers are embedded. Findings show how ACT principles fit as a stand-alone model or in conjunction with other grief theories. As such, important insights were obtained on how ACT training and MGMW can assist in the delivery of bereavement support.

There are some limitations worthy of consideration. For instance, there was a distinct difference in engagement between organisation one and two as the study progressed. Through consultation, it seemed there were unexpected extenuating circumstances in some departments at one organisation who were operating at reduced capacity which is thought to have impacted participant engagement. Additionally, the process of how clients were matched to volunteers was based on local organisational procedures therefore were not rigorously detailed.

Participants agreed to take part in online training therefore, these findings may not apply to other volunteers. Training was delivered via Microsoft Teams; live participant attendance was automatically recorded. However, there was no means to verify those who stated they viewed training sessions retrospectively. That said, there was no reason to question

participant integrity or truthfulness. Additionally, ACT principles were often used alongside other bereavement theories and there was no measure of fidelity in how ACT was used in practice or if perceived improvements were as a result of ACT or other bereavement approaches.

It is also important to point out that data excludes the views and opinions of those who did not take part in focus groups or interviews therefore, findings may not have captured wider viewpoints in this study. And finally, participant diversity and findings being transferrable in context and setting could be called into question as all were of white ethnicity and worked in the UK.

**Implications for bereavement support and conclusion**

Participants in this study clearly articulated programme benefits, and the adaptable application ACT principles afforded to those offering bereavement support. Based on participant insights and recommendations, we propose an expansion of grief scenarios to meet the multifactorial grieving process. This includes being sensitive to ethnicity, culture and linguistic factors.

Further research is needed to explore approaches to a national roll-out of the training programme to address unmet need for bereavement support. Such a large-scale venture will require campaigns to highlight this public health issue, paying particular attention to issues of safety and quality. For instance, if MGMW was universally available, with sufficient volunteer staff across multiple organisations who had the basic or intermediate ACT training, then the landscape could support triage and access to bereavement support at the appropriate three component levels; a configuration of self-help, guided self-help, or professional complex help using MGMW as a support mechanism.

Study findings provide preliminary evidence for the acceptability of ACT-based bereavement training for volunteers and the delivery of ACT to bereaved individuals.

**Supporting information**

**S1 File. ACT training handout – session 1.**
(PDF)

**S2 File. ACT training handout – session 2.**
(PDF)

**S3 File. ACT training handout – session 3.**
(PDF)

**S4 File. ACT training handout – session 4.**
(PDF)

**S5 File. Focus group/interview Topic Guide.**
(DOCX)

**S6 File. Participant information sheet.**
(DOCX)

**S7 File. Logic Model.**
(PPTX)

**S8 File. Logic model manuscript: A logic model to guide "My Grief My Way": An intervention development study for a digital psychological support package for unmet bereavement support needs.** (in preparation).
(DOCX)

## Acknowledgments

We are sincerely grateful to all bereavement support volunteers for their dedication and enthusiasm during the study, without whom we could not explore the potential benefits of ACT online training for bereavement support.

## Author contributions

**Conceptualization:** David Gillanders, Brooke Swash, Juliet Spiller, Emily Harrop, Lucy Selman, Anne Finucane.

**Data curation:** Anne Canny.

**Formal analysis:** Anne Canny, Tamzin Burnett, Brooke Swash, Anne Finucane.

**Funding acquisition:** David Gillanders, Anne Finucane.

**Investigation:** Anne Canny, David Gillanders, Tamzin Burnett.

**Methodology:** Anne Canny, David Gillanders, Anne Finucane.

**Project administration:** Anne Canny, David Gillanders, Anne Finucane.

**Resources:** David Gillanders, Anne Finucane.

**Supervision:** David Gillanders.

**Validation:** David Gillanders, Anne Finucane.

**Writing – review & editing:** David Gillanders, Tamzin Burnett, Brooke Swash, Juliet Spiller, Emily Harrop, Lucy Selman, Nicola Reed, Anne Finucane.

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
