## [Decision Letter · Decision Letter 0]

29 Aug 2025

Dear Dr. Canny,

Thank you for submitting your manuscript to PLOS ONE. After careful consideration, we feel that it has merit but does not fully meet PLOS ONE’s publication criteria as it currently stands. Therefore, we invite you to submit a revised version of the manuscript that addresses the points raised during the review process.

**ACADEMIC EDITOR: **

We look forward to receiving your revised manuscript.

Kind regards,

JONATHAN BAYUO, PhD

Academic Editor

PLOS ONE

Journal Requirements:

“This research was funded by a Research Project Grant from Marie Curie, Ref: MC-21-808. AF is funded by a Marie Curie Research Fellowship.”        

5. In the online submission form you indicate that your data is not available for proprietary reasons and have provided a contact point for accessing this data. Please note that your current contact point is a co-author on this manuscript. According to our Data Policy, the contact point must not be an author on the manuscript and must be an institutional contact, ideally not an individual. Please revise your data statement to a non-author institutional point of contact, such as a data access or ethics committee, and send this to us via return email. Please also include contact information for the third party organization, and please include the full citation of where the data can be found.

7. Please amend your manuscript to include your abstract after the title page.

8. We note that Supporting Information S1-S4 in your submission contain copyrighted images. All PLOS content is published under the Creative Commons Attribution License (CC BY 4.0), which means that the manuscript, images, and Supporting Information files will be freely available online, and any third party is permitted to access, download, copy, distribute, and use these materials in any way, even commercially, with proper attribution. For more information, see our copyright guidelines: http://journals.plos.org/plosone/s/licenses-and-copyright.

a. You may seek permission from the original copyright holder of Supporting Information S1-S4 to publish the content specifically under the CC BY 4.0 license.

9. We note that Supporting Information S2 includes an image of a [patient / participant / in the study].

10.If the reviewer comments include a recommendation to cite specific previously published works, please review and evaluate these publications to determine whether they are relevant and should be cited. There is no requirement to cite these works unless the editor has indicated otherwise. 

Reviewers' comments:

Reviewer's Responses to Questions

**Comments to the Author**

1. Is the manuscript technically sound, and do the data support the conclusions?

Reviewer #1: Yes

Reviewer #2: Yes

2. Has the statistical analysis been performed appropriately and rigorously?

Reviewer #1: I Don't Know

Reviewer #2: I Don't Know

3. Have the authors made all data underlying the findings in their manuscript fully available?

Reviewer #1: Yes

Reviewer #2: Yes

4. Is the manuscript presented in an intelligible fashion and written in standard English?

Reviewer #1: Yes

Reviewer #2: Yes

Reviewer #1: Noticed Acceptance and Commitment Therapy-based training and Acceptance and Commitment Training are used interchangeably. For example, Title and abstract.

Findings show how ACT principles fit as a stand-alone model or in conjunction with other grief theories. Are the Findings statistically significant?

Limitation: N is low, and the Bereavement Support Volunteers are predominantly females.

Does support received over the telephone, online, or in person affect the outcome?

Reason for withdrawal- Did holidays have an impact on leaving? Recruitment began on November 21, 2023, and ended on December 29, 2023.

Reviewer #2: The article 'Views and experiences of bereavement support volunteers on an online Acceptance

and Commitment Therapy-based (ACT) training programme.' The article reports that it is one of the first studies to explore the perspectives of bereavement support volunteers on an ACT-based online intervention, filling a significant gap in bereavement support research. Volunteers found ACT principles congruent with existing therapeutic approaches, supporting integration into diverse practices. Limitations such as diversity of the sample, selection bias, drop-outs. Please also add the following limitations:

1. The process of how clients were matched to volunteers was not rigorously detailed. This could introduce variability in the application of ACT strategies if matching was inconsistent in client needs or volunteer experience.

2. The study solely reports volunteer perceptions without incorporating any direct client-reported outcomes or experiences. Without client feedback or quantitative data (e.g., grief symptom reduction), the efficacy of the intervention remains speculative.

3. There was no systematic evaluation of volunteers' understanding of ACT before and after training. Lacking objective skill assessments (e.g., via role-plays, observed sessions) weakens conclusions on training effectiveness.

4. Potential Hawthorne effect.

**Do you want your identity to be public for this peer review?** For information about this choice, including consent withdrawal, please see our Privacy Policy

Reviewer #1: **Yes: ** Anoop Narahari

Reviewer #2: **Yes: ** Nikhil Tondehal

---

## [Author Response · Author response to Decision Letter 1]

7 Oct 2025

Thank you for your comments which we feel we have answered in context of our study. We have made adjustments where necessary and feel the paper is clarified and further enriched as a result. Thank you once again.

---

## [Decision Letter · Decision Letter 1]

6 Nov 2025

A qualitative study of bereavement support volunteers’ views and experiences on an online Acceptance and Commitment Therapy-based (ACT) training programme.

PONE-D-25-29547R1

Dear Dr. Canny,

We’re pleased to inform you that your manuscript has been judged scientifically suitable for publication and will be formally accepted for publication once it meets all outstanding technical requirements.

Kind regards,

JONATHAN BAYUO, PhD

Academic Editor

PLOS ONE

Additional Editor Comments (optional):

Many thanks for addressing the comments raised.

Reviewers' comments:

Reviewer's Responses to Questions

**Comments to the Author**

Reviewer #1: All comments have been addressed

2. Is the manuscript technically sound, and do the data support the conclusions?

Reviewer #1: Yes

3. Has the statistical analysis been performed appropriately and rigorously?

Reviewer #1: N/A

4. Have the authors made all data underlying the findings in their manuscript fully available?

Reviewer #1: Yes

5. Is the manuscript presented in an intelligible fashion and written in standard English?

Reviewer #1: Yes

Reviewer #1: All of my comments have been addressed. This is a qualitative study therefore no

statistical analyses were conducted. Title was amended.

**Do you want your identity to be public for this peer review?** For information about this choice, including consent withdrawal, please see our Privacy Policy

Reviewer #1: **Yes: ** Anoop Narahari

---

## [Editor Report · Acceptance letter]

PONE-D-25-29547R1

PLOS ONE

Dear Dr. Canny,

I'm pleased to inform you that your manuscript has been deemed suitable for publication in PLOS ONE. Congratulations! Your manuscript is now being handed over to our production team.

Kind regards,

on behalf of

Dr. JONATHAN BAYUO

Academic Editor

PLOS ONE